# Nucleation of Helium in Liquid Lithium at 843 K and High Pressures

**DOI:** 10.3390/ma15082866

**Published:** 2022-04-13

**Authors:** Jordi Martí, Ferran Mazzanti, Grigori E. Astrakharchik, Lluís Batet, Laura Portos-Amill, Borja Pedreño

**Affiliations:** 1Department of Physics, Polytechnic University of Catalonia-Barcelona Tech, 08034 Barcelona, Spain; ferran.mazzanti@upc.edu (F.M.); grigori.astrakharchik@upc.edu (G.E.A.); lluis.batet@upc.edu (L.B.); 2Barcelona School of Telecommunications Engineering, Polytechnic University of Catalonia-Barcelona Tech, 08034 Barcelona, Spain; lauraportosamill@gmail.com (L.P.-A.); borpm1996@gmail.com (B.P.)

**Keywords:** nucleation, breeding blankets, fusion reactors, helium–lithium mixtures

## Abstract

Fusion energy stands out as a promising alternative for a future decarbonised energy system. In order to be sustainable, future fusion nuclear reactors will have to produce their own tritium. In the so-called breeding blanket of a reactor, the neutron bombardment of lithium will produce the desired tritium, but also helium, which can trigger nucleation mechanisms owing to the very low solubility of helium in liquid metals. An understanding of the underlying microscopic processes is important for improving the efficiency, sustainability and reliability of the fusion energy conversion process. The spontaneous creation of helium droplets or bubbles in the liquid metal used as breeding material in some designs may be a serious issue for the performance of the breeding blankets. This phenomenon has yet to be fully studied and understood. This work aims to provide some insight on the behaviour of lithium and helium mixtures at experimentally corresponding operating conditions (843 K and pressures between $10^{8}$ and 10${}^{10}$ Pa). We report a microscopic study of the thermodynamic, structural and dynamical properties of lithium–helium mixtures, as a first step to the simulation of the environment in a nuclear fusion power plant. We introduce a new microscopic model devised to describe the formation of helium droplets in the thermodynamic range considered. Our model predicts the formation of helium droplets at pressures around 10${}^{9}$ Pa, with radii between 1 and 2 Å. The diffusion coefficient of lithium (2 Å${}^{2}$/ps) is in excellent agreement with reference experimental data, whereas the diffusion coefficient of helium is in the range of 1 Å${}^{2}$/ps and tends to decrease as pressure increases.

## 1. Introduction

Within the framework of future energy supply, with the constraints posed by the need of electrification of the final energy demand, and the quest for more sustainable power generation methods in order to achieve a decarbonised electricity system; nuclear fusion energy stands out as a promising alternative. The fusion reaction that results in being the most convenient in the present state of technological development is:(1)$$D+T\to {\;}^{4}\mathrm{He}+n+17.6\;\mathrm{MeV},$$
where ‘D’ stands for deuterium, ‘T’ for tritium and ‘n’ for a free neutron and where helium is a by-product [1]. Deuterium is abundant in water, but tritium ($t_{1/2}$ = 12.3 year) must be artificially created. Therefore, in order for fusion energy to be sustainable, it is necessary that tritium be produced in the reactor itself. Tritium will be generated by means of the reactions of neutrons escaping from the plasma with lithium in the so-called breeding blankets (see, for instance, [2] for an overview of these relevant components in DEMO, a demonstration power plant contemplated in the European Roadmap to Fusion). Breeding blankets (BB) will perform two additional functions besides producing tritium: extraction of fusion heat and shielding the magnets (superconducting coils) from the radiation escaping the plasma.

Lithium has two natural isotopes ${}^{6}$Li (abundance 7.5%) and ${}^{7}$Li (92.5%), both producing tritium when capturing a neutron [3]:(2)$$n+{\;}^{6}\mathrm{Li}\to T+{\;}^{4}\mathrm{He}+4.78\;\mathrm{MeV}$$
(3)$$n+{\;}^{7}\mathrm{Li}\to T+{\;}^{4}\mathrm{He}+n-2.47\;\mathrm{MeV}$$

Tritium self-sufficiency will require a certain neutron multiplication in order to close the fuel cycle with a net gain so that the so-called tritium breeding ratio is greater than 1. In order to fulfil their functions, some BB designs feature solid (ceramic) breeders cooled by helium, while others rely on a liquid metal (LM) cooled by helium or water. The LM BB designs are considering the use of lithium–lead eutectic (LLE) [2,4,5]. Besides ${}^{7}$Li, lead will provide some fast neutron multiplication (neutrons hit the walls of the reaction chamber with energies bigger than 14 MeV). As shown in Equations (Equation 2) and (Equation 3), He is produced mol-to-mol along with T. However, He is practically insoluble in the liquid metal (Henry’s constant for helium in Li at 843 K would be around 7 × 10${}^{-14}$ Pa${}^{-1}$ atomic fraction; for LLE, it is estimated to be lower [6]). Tritium self-sufficiency requirement is thus linked to a possible super-saturation of helium in the liquid metal and, consequently, to a possible nucleation of helium in the form of bubbles. This phenomenon may have a great impact in the performance of the BB: changes in the magnetohydrodynamic flow, affectation of the heat transfer, and changes in the tritium migration mechanisms. Other systems that could be affected by helium nucleation are, for instance, free-surface Li first wall concepts [7,8] and the Li jet targets in the future International Fusion Materials Irradiation Facility [9].

In the quest for tools to model the effect of the undesired helium bubbles being formed in the blanket walls of a nuclear fusion plant, helium nucleation models must be developed. Thus far, no experiments exist allowing for validating such models. The low solubility of He in LM makes computer simulations extremely expensive when trying to capture the onset of nucleation at the design operational pressures and temperatures of BB. Indeed, a rough estimation based on the Gibbs’ Classical Nucleation Theory [10,11] (CNT) can be done using parameters from Ref. [12]. In order to have a stable bubble, a critical size must be achieved when the internal energy of the bubble is able to overcome the energy needed to create the surface around it. The smaller is the critical size, the higher is the supersaturation level needed to achieve it.

To highlight this point, we report in Figure 1 a graphical representation of the work of formation (Gibbs free energy) of a cluster of radius $r_{c}$ (see [12]):(4)$$\Delta G_{tot}=\Delta G_{surf.}+\Delta G_{vol.}=4\pi \;r_{c}^{2}\sigma +\frac{4}{3}\pi \;r_{c}^{3}\Delta g_{vol.},$$
where σ is the surface tension and $\Delta g_{vol.}$ is the driving force for nucleation per unit volume of the new phase i.e., the Gibbs free energy difference between the cluster and the dissolved states of one He atom per unit volume. According to CNT, it can be expressed as [13,14]:(5)$$\Delta g_{vol.}=\frac{-k_{B}T}{v_{0}}\ln\psi ,$$
where $k_{B}$ is the Boltzmann constant, *T* is the liquid metal bulk temperature, $v_{0}$ is the volume of one He atom in the cluster and ψ is the supersaturation ratio, relating the actual He concentration to the saturation concentration.

Given a fixed concentration of helium in the solvent, when the solubility increases (i.e., ψ decreases), the critical bubble size (i.e., the radius at which the total Gibbs free energy is maximum) is larger (in the example of Figure 1
ψ equals 2 for a critical size of 110 atoms of helium). If the solubility is lower, the critical bubble size is smaller (in the example of Figure 1
ψ equals 6 for a critical size of 42 atoms of helium). Thus, in order to have a stable bubble, a critical size must be achieved when the internal energy in the bubble is able to overcome the energy needed to create the surface around it. The smaller the critical size (around 40 atoms in the example above), the higher the supersaturation level needed to achieve it. Consequently, a simulation involving 40 atoms of helium at 843 K and 1 bar would require almost 10${}^{9}$ atoms of lithium to be in those conditions. However, at 100 GPa only, around 1000 atoms of lithium would be needed.

The complexity of the chemistry of the LLE system, with bred tritium and helium, including possible interactions between all types of atoms and the possibility of the formation of molecules (LiT when the eutectic composition is not well adjusted and Li${}_{2}$ and T${}_{2}$ in the gas phase), makes it unaffordable to try to model the interactions between all possible species at once, when the nucleation mechanism in this case has not been fully captured by models yet. For these reasons, the present work focuses on the simulation of He-Li mixtures at high pressures as a first step towards the simulation of the Li-Pb-He mixtures at low pressure. Our main goal is to capture the onset of the nucleation at a qualitative level, in order to advance towards the full modelling of the phenomenon. We describe a mixture of helium and lithium atoms in the bulk, developing a microscopic model that is able to reproduce the helium–lithium mixture instability towards nucleation of helium droplets. We thus acquire valuable structural and dynamical data from classical simulations, showing remarkable good agreement with radial distribution functions of lithium as well as with its self-diffusion coefficient, compared to data from experimental and computational sources. Full verification of the model is limited by the current lack of available data on Li-He mixtures at high temperatures and pressures.

In particular, Li-Li, He-He and He-Li pair interactions are fed as an input to both classical Monte Carlo (MC) and molecular dynamics (MD) simulations. We find thermodynamic, structural and dynamic properties of lithium and helium mixtures at high temperatures and in a wide range of pressures between 0.1 and 10 GPa. Both MC and MD computational techniques have been previously proven to provide reliable predictions for a wide variety of classical and quantum atomic and molecular systems, ranging from pure quantum systems including hydrogen and helium [16,17,18,19] to classical molecular liquids such as water, in solution [20,21,22] and at interfaces [23,24], and to highly complex biosystems such as proteins or membranes [25,26]. MC and MD can be the source of mixed methods such as transition path sampling [27,28], which is able to describe the free energy hypersurface of a given statistical process without the previous knowledge of the reaction coordinates. We calculate and report thermodynamic properties such as the average internal energy as a function of pressure. To quantify the spatial and dynamical structure, we calculate atomic pair distribution functions, structure factors, mean squared displacements and velocity autocorrelation functions in order to obtain atomic spectra. We also can obtain the diffusion coefficients of lithium and helium at different pressures as well as the spectral densities of He and Li, reporting information on their main translation and vibration modes.

## 2. Methods

### 2.1. Microscopic Model

We base our simulations on a microscopic model Hamiltonian describing a mixture of $N_{\mathrm{Li}}$ lithium and $N_{\mathrm{He}}$ helium atoms, which are taken to be point-like particles of mass $m_{\mathrm{Li}}$ and $m_{\mathrm{He}}$, respectively. In order to reproduce the experimental conditions, we only consider situations where $N_{\mathrm{Li}}\gg N_{\mathrm{He}}$. Each species is characterised by particle coordinates and velocities $\left\{r_{\mathrm{Li},i},v_{\mathrm{Li},i}\right\}$ and $\left\{r_{\mathrm{He},j},v_{\mathrm{He},j}\right\}$, with *i* and *j* spanning the ranges $1,\ldots ,N_{\mathrm{Li}}$ and $1,\ldots ,N_{\mathrm{He}}$, respectively. The Hamiltonian of the system is then written as
(6)$$\begin{array}{ccc} H & = & \frac{1}{2}\sum_{i=1}^{N_{\mathrm{Li}}}m_{\mathrm{Li}}v_{\mathrm{Li},i}^{2}+\frac{1}{2}\sum_{i=1}^{N_{\mathrm{He}}}m_{\mathrm{He}}v_{\mathrm{He},i}^{2}\\ & + & \sum_{i<j}^{N_{\mathrm{Li}}}V_{\text{Li-Li}}\left(\right|r_{\mathrm{Li},i}-r_{\mathrm{Li},j}\left|\right)+\sum_{i<j}^{N_{\mathrm{He}}}V_{\text{He-He}}\left(\right|r_{\mathrm{He},i}-r_{\mathrm{He},j}\left|\right)+\sum_{i=1}^{N_{\mathrm{Li}}}\sum_{j=1}^{N_{\mathrm{He}}}V_{\text{Li-He}}\left(\right|r_{\mathrm{Li},i}-r_{\mathrm{He},j}\left|\right)\;, \end{array}$$
where the first two terms describe the kinetic energy, while the last three terms account for the intra- and inter-species interaction, respectively. Periodic boundary conditions are applied in order to minimise the finite-size effects and approximate better the properties of a large system. The typical simulation cage is a square box of length around 29 Åfor the reference pressure of 1 GPa. At lower pressure setups, box lengths are larger than 40 Å.

A crucial point of our model is an appropriate choice of the pair interaction potentials. For lithium–lithium interactions (Equation (Equation 7)), we rely on the model proposed by Canales et al. in Refs. [29,30], whereas the remaining interactions are a novelty of the present work. The Li-Li pair potential V(r) is modelled as [29,30]
(7)$$V_{\text{Li-Li}}\left(r\right)=Ar^{-12}+B\exp Cr\cdot \cos D(r-E),$$
where *r* is the distance between the two atoms in Angströms and the potential coefficients are $A=2.22125\times 10^{7}\;$K Å${\;}^{12}$, B=41828.9 K, C=−1.20145 Å${}^{-1}$, D=1.84959 Å${}^{-1}$, E=5.03762 Å. This potential is shown in Figure 2, featuring strong short-distance repulsion caused by Pauli exclusion due to overlapping electron orbitals, a highly non-monotonic behaviour at the distances around the van der Waals radii and an attractive long-range tail. In particular, the characteristic length of the potential corresponds to the smallest distance at which the interaction changes sign, $V_{\text{Li-Li}}\left(\sigma_{\text{Li-Li}}\right)=0$, and is equal to $\sigma_{\text{Li-Li}}=2.5668$ Å. The characteristic energy scale is defined by the depth of the first minimum, equal to $\epsilon_{\text{Li-Li}}\equiv V_{\text{Li-Li}}\left(3.06\right)=-887.9$ K.

The helium–helium interaction is considered to be of the Lennard–Jones (LJ) type, and it is parameterised to accurately describe the system at the temperatures and pressures of interest which are well beyond ambient conditions. From preliminary simulations, we have found that the Aziz II potential model [31], which is known to provide an excellent description of superfluid liquid helium at temperatures close to absolute zero and moderate pressures around saturation density, does not apply quite well at temperatures as high as 843 K and pressures in the GPa regime considered in the present study when combined with the Li-Li model given above (Equation (Equation 7)). Instead, we retain the same width $\sigma_{\text{He-He}}=2.556$ Å but treat the potential depth $\epsilon_{\text{He-He}}$ as a free adjustable parameter. In this work, we have found that, in order to be able to reproduce the nucleation process, the depth must be increased to the typical values of the Li-Li potential (Equation (Equation 7)). In order to test the influence of this interaction parameter, we considered two different values of the potential depth, $\epsilon_{\text{He-He}}=-1200$ K and $\epsilon_{\text{He-He}}=-800$ K, henceforth referred to as “model 1” and “model 2”, respectively.

Finally, the helium–lithium interaction is modelled by a truncated Lennard–Jones potential at short distances, namely a “hard" wall where we have basically eliminated the attractive part, with characteristic parameters given by the Lorentz–Berthelot rules obtained from the corresponding Li-Li and He-He values, and a cutoff beyond that point. This results in $\sigma_{\text{Li-He}}=2.5615$ Å and $\epsilon_{\text{Li-He}}=-1032.2$ K. The potential model is given by:(8)$$\begin{array}{ccc} V_{\text{Li-He}}\left(r\right) & = & 4\epsilon_{\text{Li-He}}\left[{\left(\frac{\sigma }{r}\right)}^{12}-{\left(\frac{\sigma }{r}\right)}^{6}\right],\;r\leq \sigma_{\text{Li-He}}\\ & = & 0,\;r>\sigma_{\text{Li-He}} \end{array}$$

The four considered pairwise interactions are shown in Figure 2. In a very recent work [32], it has been reported that specific interatomic potentials based on Daw–Baskes and Finnis–Sinclair formalisms are able to describe the formation of helium bubbles in a palladium tritide lattice at temperatures of the order of 400 K and pressures in the range of 0.1 to 2.2 GPa. Furthermore, the formation of helium bubbles in tungsten was also reproduced using purely repulsive He-W interaction potentials in cluster simulations [33], models rather close to the ones presented in this work.

### 2.2. Monte Carlo and Molecular Dynamics Methods

We rely on MC and MD methods to perform a series of computer simulations of the system. Both methods use the microscopic model introduced in the previous section to describe the interactions between the atoms as an input.

The Monte Carlo method has been used to obtain the equilibrium properties at fixed pressure *P*, particle number *N* and temperature *T*. Calculations are performed starting from the microscopic Hamiltonian of Equation (Equation 6), using it to define the probability of a state with energy *E* according to the Maxwell–Boltzmann distribution, $p=\exp(-E/k_{B}T)$, which is sampled using the standard Metropolis algorithm. Once the system has been equilibrated, we perform simulations to estimate quantities of interest such as the energy per particle and the volume, as well as correlation functions such as the pair distribution function and the low-momentum static structure factor. An advantage of the MC method is that it only uses the particle positions, in contrast to MD where their velocities have also to be sampled. This halves the number of microscopic variables to estimate, thus reducing the phase space and making the exploration more efficient. This, however, comes at a price: since Monte Carlo can only sample equilibrium configurations, it is not able to provide information about the time-dependent properties, in contrast to MD where the simulation propagates in real time.

In molecular dynamics, the force fields are also obtained from the model in Equation (Equation 6) and the corresponding Newton’s equations of motion, which are integrated numerically using a standard leap-frog Verlet procedure [34]. In each simulation, we fix the number of particles *N* and the pressure *P*, while the volume is adjusted accordingly. In addition to the energetic and structural properties obtained also in MC, MD provides access to time-dependent quantities such as the diffusion coefficient, velocity autocorrelation functions and spectral densities. As a stringent test of self-consistency, strict agreement between the common quantities sampled in MC and MD has to be obtained, which requires the proper thermalization and averaging in both methods.

## 3. Results and Discussion

In all cases, a homogeneous mixture of helium and lithium has been considered as the starting point of the simulations. The concentration of helium has been set to ∼0.04 for a total of 40 helium atoms dissolved in a sea of 960 lithium atoms. The main results for the thermodynamic quantities of interest obtained in both MC and MD are summarised in Table 1. Additional simulations at intermediate pressures (0.3 and 0.4 GPa for instance) have also been considered in several other sections of the manuscript.

As a starting point of our analysis, we have obtained the average internal energies and pressures, as reported in Table 1. In the MC simulations, the system was initially allowed to equilibrate for a total of $10^{7}$ MC random movements. The statistics was collected over the subsequent $10^{8}$ random steps. In the case of MD, we employed a total time of about 50 ps for the equilibration of each system and later on we collected MD trajectories 200 ps long in all cases to compute meaningful physical properties. In both MC and MD, the statistical errors were less than 1% in all reported quantities. The state of lowest internal energy is found at the pressure of 1 GPa. As an additional test, and in order to explore the influence of the helium concentration on the total energy of the system, we report energies as a function of the relative helium concentrations in Figure 3. We define the relative helium concentration *p* as
(9)$$p=\frac{n_{\mathrm{Li}}-n_{\mathrm{He}}}{n_{\mathrm{Li}}+n_{\mathrm{He}}}$$
and observe a monotonic behaviour at the lowest pressure (0.1 GPa), while it becomes non-monotonic for the second pressure (0.3 GPa). This might be an indication of a different qualitative phase coexistence for the two selected pressures. We report further information about this aspect in the following sections.

It has been reported in Ref. [35] that the solubility of helium in lithium is ∼$5\times 10^{-7}\;\mathrm{mol}/(L\cdot \mathrm{bar})$ at the pressure of 2.38 bar for temperatures in the range of 922–1144 K, which is in agreement with Henry’s law. At the same time, one should keep in mind that Henry’s law is based on the assumption that the system behaves as an ideal gas and describes the overall incompressibility of liquid metals. Such a low solubility value means that, when applied to the typical conditions of our simulations (pressure of 1 GPa inside a volume of 24.26 nm${}^{3}$, see Figure 4), only around 0.06 helium atoms would be able to dissolve.

This explains why simulations performed at 843 K and high pressures (helium inverse densities are within the cubic nanometer range) are able to capture the phenomenon of helium nucleation, where helium cannot dissolve. Conversely, at lower pressures, helium is able to dissolve in lithium. Furthermore, at the very high densities and pressures inside the projected nuclear fusion facilities, Henry’s law is rather unlikely to apply due to large deviations from the ideal gas behaviour. Figure 3 reports dependence of the specific volume (defined as the inverse of the density $n^{-1}$) on the pressure. One can see that the pressure rapidly increases as the specific volume is diminished, and that a plateau is reached once the mean interparticle distance $n^{-1/3}$ becomes comparable to the hard-core size σ. At that point, the pressure can increase without a significant change in the density resulting in a vanishing compressibility
(10)$$\kappa =-\frac{1}{V}\frac{\partial V}{\partial P}=-\frac{1}{n^{-1}}\frac{\partial n^{-1}}{\partial P}\to 0,\;P\to \infty \;.$$

On the other hand, the observed asymptotic incompressibility of lithium is in overall agreement with Henry’s law.

### 3.1. Structure: Pair Distribution Function and Structure Factor

In order to quantify the spatial correlations and visualise the structure of helium droplets, we evaluate the pair distribution function g(r) (RDF) in the simulated mixture of 960 Li and 40 He atoms at 843 K. The RDF quantifies the probability of finding two atoms of species α and β at a distance *r*,
(11)$$g_{\alpha ,\beta }\left(r\right)=\frac{1}{N_{\alpha }N_{\beta }}\sum_{i=1}^{N_{\alpha }}\sum_{j=1}^{N_{\beta }}\langle \delta (|r{|}_{ij}-r)\rangle \;,$$
where 〈⋯〉 denotes a thermal average. Being a two-particle correlator, the RDF is capable of capturing a translationally invariant ordering and is therefore suitable to identify droplet formation independently of its center of mass position. Typical RDF functions for Li-Li, Li-He and He-He pairs are shown in Figure 5 for different pressures. The shape of the Li-Li pair distribution functions is characteristic of a liquid at equilibrium. The Li-Li RDFs are hardly affected by the presence of a small concentration of helium atoms, as it can be seen in comparison with the behaviour of pure lithium at 1 GPa and the same temperature, taken from Ref. [30]. One might also note that a change of two orders of magnitude in the pressure does not significantly affect the overall shape of $g_{\text{Li-Li}}\left(r\right)$. The short-range region is voided due to the steep hard-core potential. High-amplitude oscillations appear at separations comparable to the mean interparticle distance and witness strong correlations in the liquid that can be interpreted as shell effects.

At large distances, the pair distribution function approaches a constant value, thus confirming that lithium atoms are homogeneously filling the whole space. The situation is drastically different in the He-He RDFs, as they vanish at large distances as seen in Figure 5c. While at low pressure, $g_{\text{He-He}}$ still shows a long-range plateau; this is not the case for large pressure where the RDF strongly decreases. This implies that helium atoms bunch up close to each other, thus forming droplets. In this way, helium atoms form a miscible mixture on a lithium background at low pressure but have a tendency to phase separate at large pressures, splitting the system into pure lithium and helium phases. This scenario is further supported by the massive increase in the height of the first and subsequent shells in a He droplet. The droplet size can be roughly estimated as the difference between the distance at which $g_{\text{He-He}}\left(r\right)$ significantly decays (taken from the first minimum) and the position of the starting non-zero value of the RDF.

In order to verify the robustness of our analysis, we have compared the results obtained with the two different He-He potential models proposed (see Section 2) corresponding to a depth well of 800 K and 1200 K, to find only minor changes. We have observed that, when this depth is below 650 K, long-lived helium droplets are not formed and become significantly unstable in short time intervals of the order of 1 ps. From here on, the reported results correspond to model 1, as it predicts more stable helium droplets.

A set of four characteristic snapshots of the system at pressures P=0.1,0.2,1 and 10 GPa is shown in Figure 6 to illustrate the tendency of the system to form helium droplets when the pressure is increased above approximately 0.2 GPa. At the lowest pressures considered, helium is uniformly diluted in the lithium bath, showing that only small clusters of the size of a few helium units appear. This is also seen in the He-He pair distribution function, which is shown in Figure 7 for several values of *P* close to the critical transition pressure. We observe that larger helium droplets start to form at a crossover pressure around 0.175 GPa, which corresponds to phase separation (helium droplets in liquid lithium), and fully stable ones appear at 0.2 GPa. A possible explanation of this effect can be based on the behaviour of the binding (cohesive) potential energy of helium
(12)$$U_{binding}\equiv \frac{U_{\text{He-Li}}-U_{\mathrm{Li}}}{N_{\mathrm{He}}}$$
where $U_{\text{He-Li}}$ and $U_{\mathrm{Li}}$ stand for the internal energies of the mixture and pure lithium, respectively, and $N_{\mathrm{He}}$ is the total number of helium atoms, fixed to 40 in this work. The obtained results are reported in Table 2. Here, the values of $U_{\mathrm{Li}}$ have been extracted from additional simulations of pure lithium at 0.1, 1 and 10 GPa. We find that the cohesive potential energy is positive at low pressures where the lithium atoms prefer not to bind, while it becomes negative at large pressures, where the formation of large lithium droplets is observed. Thus, the stability of the helium droplets is significantly enhanced at high pressures.

A possible effect that could be expected is the formation of aggregates of lithium and helium due to van der Waals forces [36]. However, we have not observed that pairing in our simulations, as it can be seen from the Li-He RDF of Figure 5. This is probably due to the short-range repulsive He-Li interactions considered, as shown in Equation (Equation 8).

In order to further characterise the phase separation, we also report static structure factors S(k) computed from the RDF (Equation (Equation 11)) at low momenta ($k=0.216\;{Å}^{-1}$) as a function of the pressure (see, for instance, [37]):(13)$$S\left(k\right)=1+\frac{4\pi \rho_{0}}{k}\int_{0}^{\infty }dr\;r\;\left[g\left(r\right)-1\right]\sin kr,$$
where $\rho_{0}$ is the density of a pure species or the average ($\sqrt{\rho_{\mathrm{Li}}\rho_{\mathrm{He}}}$) for the Li-He pair correlation. The results are shown in Figure 8. The change in the slope of S(k) is particularly sharp in the He-He case, around the crossover pressure of 0.175 GPa, as opposed to what happens in the Li-Li and Li-He cases. It has recently been reported [38] that the sensibility of changes in the slope of the static structure factors may be a clear indication of a possible phase transition between a helium drop and a mixture of helium dissolved in lithium. The precise quantification of such a phase transition is currently evaluated in our lab, although it is out of the scope of the present work.

The radii of the helium droplets formed in our simulations are reported in Table 3 and represented in Figure 9, while the specific size depends on the number of particles in the simulation. We considered only the reference case of 40 Helium and 960 lithium atoms. One observes that the radius is largest at low pressures, thus decreasing as the pressure increases. At pressures below 1 GPa, we can fit an exponential law: $R/Å=1.965e^{-0.12P/\mathrm{GPa}}$, whereas, in the range above 1 GPa, the best fit is linear: R/Å = (1.84–0.05) *P*/GPa. This indicates a qualitatively different behaviour for *R* that is strongly dependent on the pressure.

### 3.2. Dynamics: Atomic Self-Diffusion Coefficients

Another experimentally relevant quantity is the diffusion coefficient. We obtain the mean square displacement (MSD) for both helium and lithium from the MD simulations. The value of the diffusion coefficient *D* is then computed from the slope of the steady-state MSD curve, using Einstein’s formula
(14)$$D=\frac{1}{6}\;\lim_{t\to \infty }\;\frac{d}{dt}{\langle |r\left(t\right)-r\left(0\right)|}^{2}\rangle ,$$
where r stands for the coordinate of each species. The coefficients for all simulated states are reported in Table 4. Canales et al. [30] obtained a value for the diffusion coefficient of pure lithium at 843 K (around 1 GPa) of 2.47 Å${}^{2}$/ps, whereas Jayaram et al. [39] reported 0.8 Å${}^{2}$/ps at 500 K. We get a similar value, D=2.0 Å${}^{2}$/ps at 843 K, indicating that the lithium diffusion coefficient does not change significantly from its value in the absence of helium. This is not surprising considering the low concentration of helium atoms in the regimes analysed. It is also worth noticing that a significantly higher experimental value of 45 Å${}^{2}$/ps at 523 K, reported by Nieto et al. [40] for helium injected onto the surface of a stream of flowing lithium, was obtained in a system out of equilibrium, which is a different situation from the one analysed here. This can explain the large difference of about two orders of magnitude when compared to our result, 0.833 Å${}^{2}$/ps at 843 K and 1 GPa (see Table 4). Experiments in similar systems might provide a more suitable reference to which we can compare our results. Figure 10 shows the dependence of *D* obtained in our simulations as a function of the pressure. As it can be seen, the dependence of *D* on *P* is approximately linear, with a slower diffusion at pressures above 1 GPa.

Experimental infrared spectra are usually obtained from the absorption coefficient α(ω) or the imaginary part of the frequency-dependent dielectric constant [41]. These properties are directly related to the absorption lineshape I(ω), which can also be obtained in molecular dynamics simulations [42,43]. In most cases, the physically relevant property to be computed is the so-called atomic spectral density $S_{i}\left(\omega \right)$:(15)$$S_{i}\left(\omega \right)=\int_{0}^{\infty }dt\;\langle {\vec{v}}_{i}\left(t\right){\vec{v}}_{i}\left(0\right)\rangle \;\cos\left(\omega t\right),$$
where ${\vec{v}}_{i}\left(t\right)$ is the velocity of the *i*-th atom at time *t*, while the brackets 〈⋯〉 denote an equilibrium ensemble average. In our case, we have obtained the spectral density of each atomic species separately. Generally speaking, classical molecular dynamics simulations are not able to fully reproduce experimental absorption coefficients, these being quantum properties. However, they can be used to locate the position of the spectral bands since, in the harmonic (oscillator) approximation, the classical and quantum ground state frequencies are equal.

The power spectrum describes the main vibrational modes of a molecular system, including low frequencies below 100 ps${}^{-1}$, associated with translational and rotational modes, and high frequencies of stretching and bending vibrations around and above 500 ps${}^{-1}$. The power spectra were obtained for the velocity autocorrelation functions of lithium and helium atoms and are reported in Figure 11. We find that lithium atoms have a tendency to oscillate at frequencies between 30 and 85 ps${}^{-1}$, whereas the vibrational frequency for helium atoms is between 2 and 100 ps${}^{-1}$, approximately. The fact that these peaks are found at low frequencies is consistent with a picture where the atoms can only present translational vibration modes, mainly associated with the restricted translations often referenced as the *cage effect*. These are typical of most condensed liquids and in the present case due to short-range interactions of a given lithium or helium atom with its closest neighbours [44]. As a general trend, we observe that translational modes decrease their values as pressure rises, as expected due to condensation effects.

## 4. Conclusions

In this work, we have analysed the structure and dynamics of lithium-helium mixtures with a very low He concentration as a first step towards the simulation of the typical environmental conditions in the BB of a fusion power plant. We perform classical simulations of the lithium–helium mixture using Monte Carlo and molecular dynamics methods, both yielding the same predictions at equilibrium. The Monte Carlo approach is more efficient for the calculation of thermodynamic quantities, and we employ it for the estimation of the total energy and pressure, along with some of its structural properties as the pair distribution functions. In addition, molecular dynamics is used to obtain time-dependent quantities such as the diffusion coefficients, velocity autocorrelation functions and power spectra of the atoms in the mixture.

In our simulations, we have observed that lithium becomes incompressible at pressures above 2 GPa, in overall agreement with Henry’s law. At the same time, the helium solubility is too low in the range of high pressures considered so we observe the formation of helium droplets within our microscopic model. Furthermore, we also find that helium atoms are miscible in the lithium bath at low pressures.

The simulations reported in this work provide a first step towards the understanding of the phenomenon of helium nucleation in liquid lithium directly from a microscopic model. We have shown that, at high temperatures and high pressures, this can be captured by classical computer simulations at its inception if appropriate potential models are used. Independently of the initial homogeneous disposition of atoms in the system, our simulations show the formation of helium droplets systematically if the same environmental conditions are met. Dynamical properties of the mixture, such as diffusion coefficients of lithium and helium, are very well reproduced, in overall good agreement with the experimental and computational data available. Future studies would likely involve the calculation of surface tensions of the droplets and the analysis of the nucleation phenomenon on lithium–lead-helium mixtures in the range of high temperatures and pressures.

## Figures and Tables

**Figure 1 materials-15-02866-f001:**
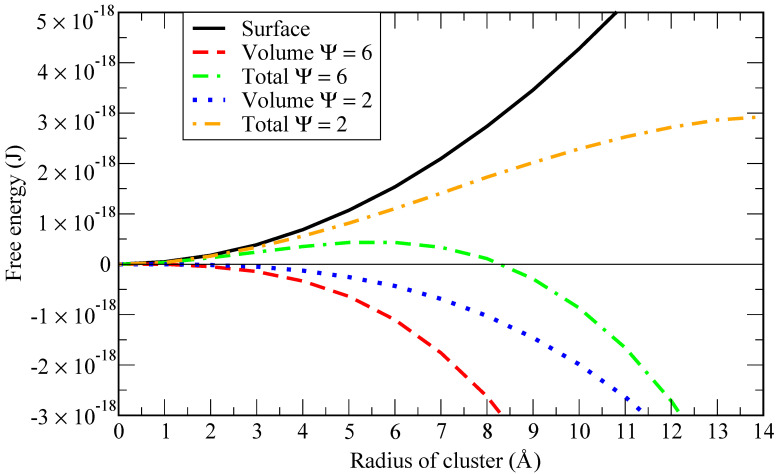
Free energy of a He cluster forming in lithium at 843 K assuming a surface tension of 0.34 N/m [15]. Using a volume of 17 Å${}^{3}$ for helium [12], the critical size is 42 atoms when supersaturation ratio ψ=6 and 110 atoms when ψ=2, where ψ is defined as the ratio between the actual helium concentration and the saturated concentration.

**Figure 2 materials-15-02866-f002:**
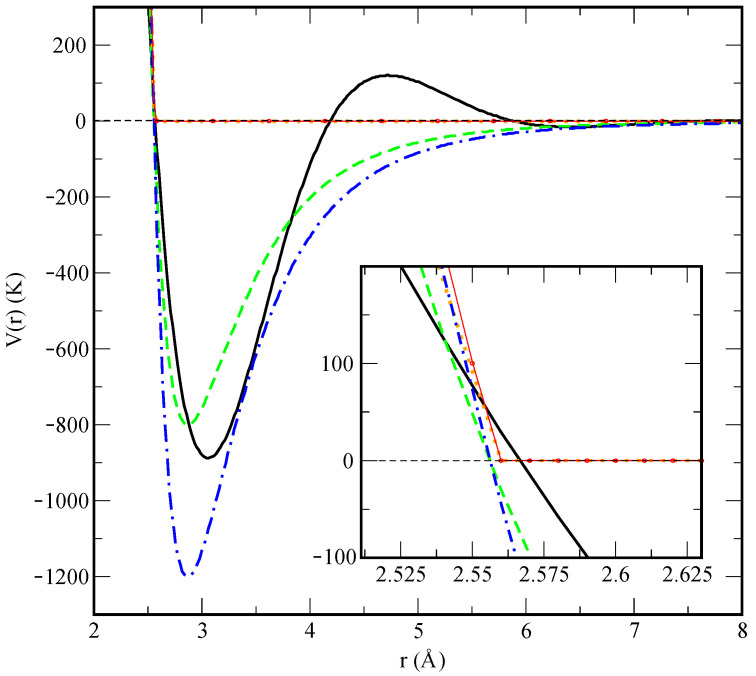
Microscopic two-body interaction potentials employed in this work. Main figure: overall view of the interaction potentials V(r). Li-Li (black line); He-He model 1 (dot-dashed blue line); He-He model 2 (dashed green line); Li-He model 1 (red circles) and Li-He model 2 (dotted orange lines). The inset represents a zoom of the “hard-wall” area at short distances.

**Figure 3 materials-15-02866-f003:**
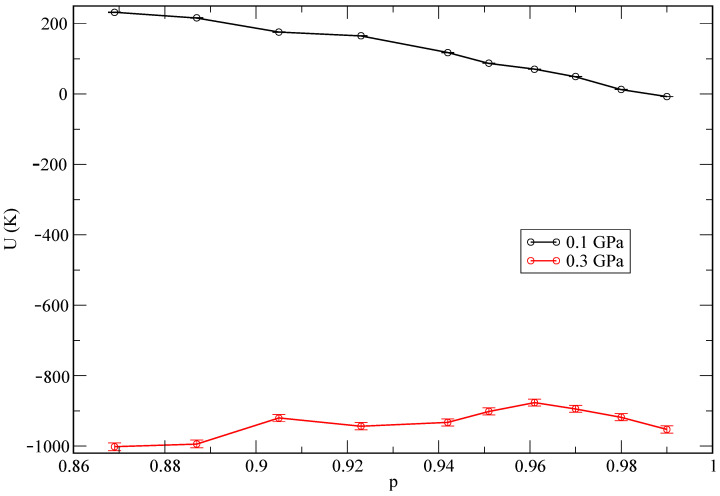
Total internal energies *U* as a function of concentration *p* for two characteristic pressures (0.1–0.3 GPa).

**Figure 4 materials-15-02866-f004:**
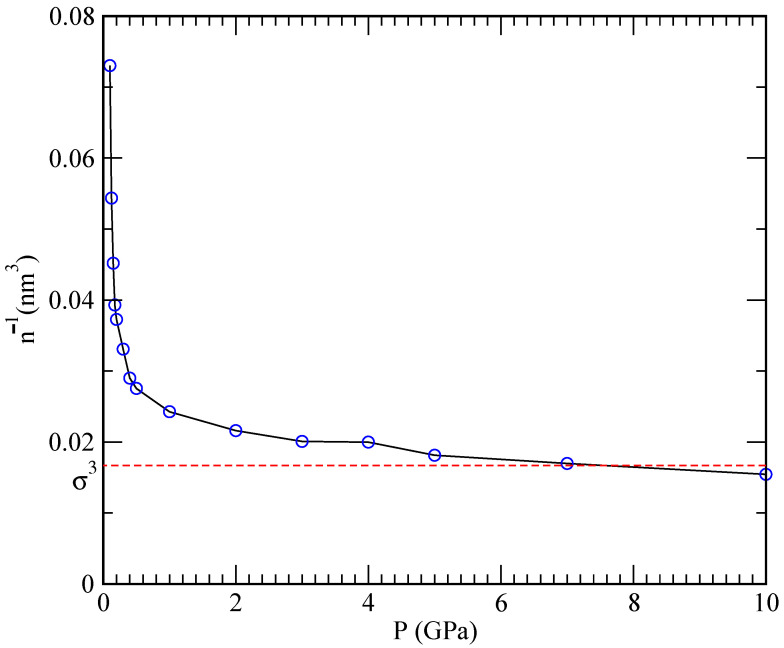
Specific volume $n^{-1}$ as a function of the pressure in a wide range of pressures *P* (0.1–10 GPa). Symbols: results of the simulation, with the error bars smaller than the symbol size. Dashed line: hard-wall volume associated with the Van der Waals radius σ of lithium and helium (both are ∼2.5 Å).

**Figure 5 materials-15-02866-f005:**
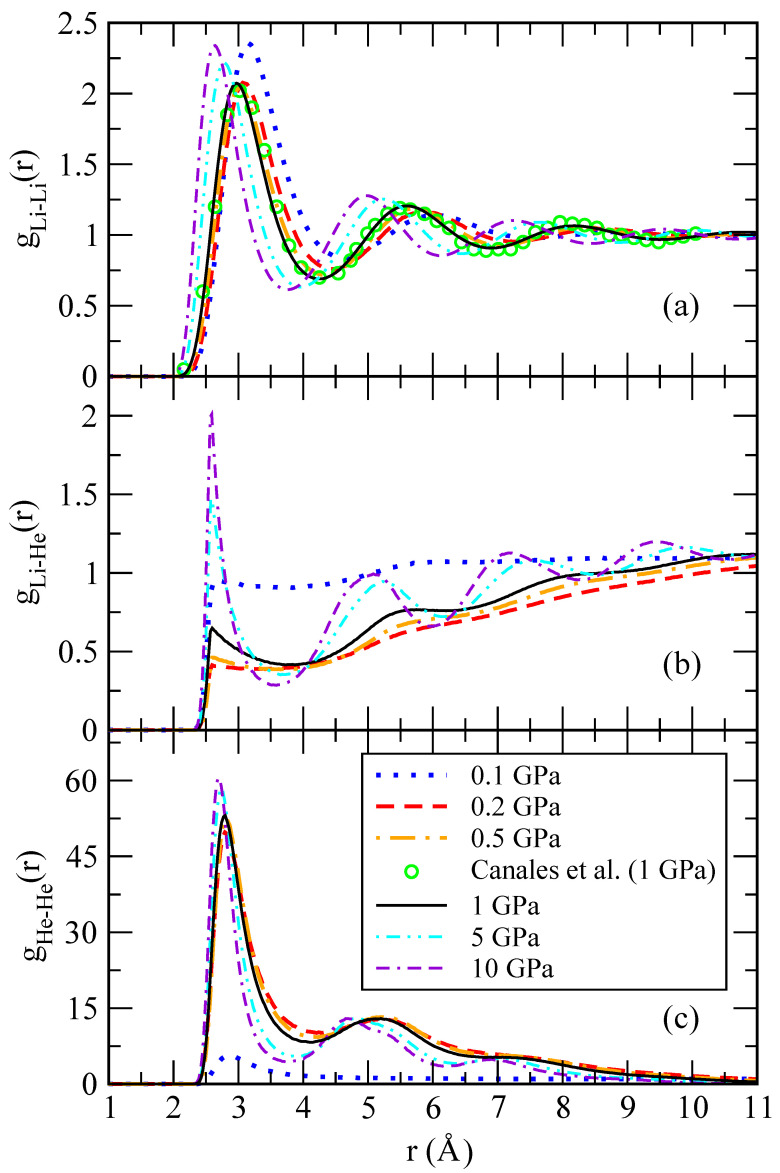
Pair distribution functions in a wide range of pressures (0.1–10 GPa) quantifying (**a**) He-He; (**b**) He-Li; and (**c**) Li-Li correlations. Green circles, single-species Li-Li data from Ref. [30]. Lines: 0.1 GPa (dotted blue); 0.2 GPa (dashed red); 0.5 GPa (dot-dashed orange); 5 GPa (dot-dot-dashed cyan); 10 GPa (dash-dash-dotted violet).

**Figure 6 materials-15-02866-f006:**
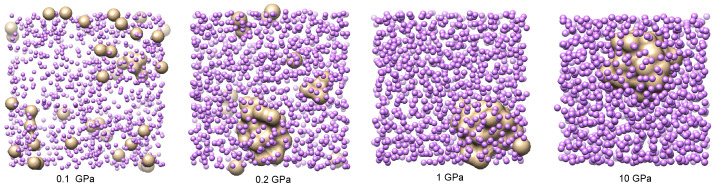
Snapshots of the He-Li mixtures at characteristic pressures: 0.1 GPa, 0.2 GPa, 1 GPa, and 10 GPa.

**Figure 7 materials-15-02866-f007:**
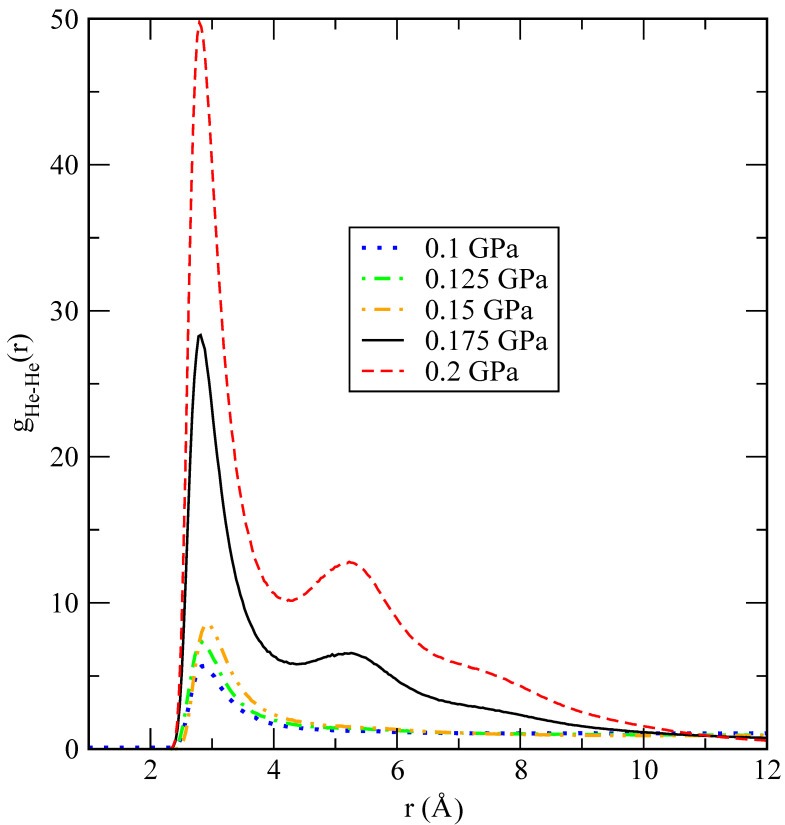
He-He pair distribution functions in the vicinity of the phase-separation transition (0.1–0.2 GPa). Lines: 0.1 GPa (dotted blue); 0.125 GPa (dot-dashed green); 0.15 GPa (dot-dot-dashed orange); 0.175 GPa (black); 0.2 GPa (dashed red).

**Figure 8 materials-15-02866-f008:**
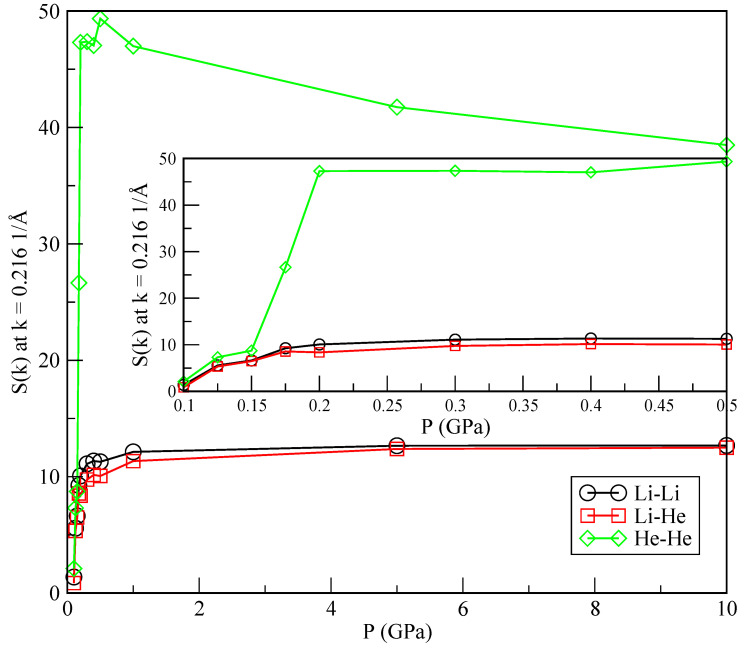
Static structure factors computed at very low momentum (k=0.2161/Å) a function of the pressure. Li-Li (black circles); Li-He (red squares); He-He (green diamonds).

**Figure 9 materials-15-02866-f009:**
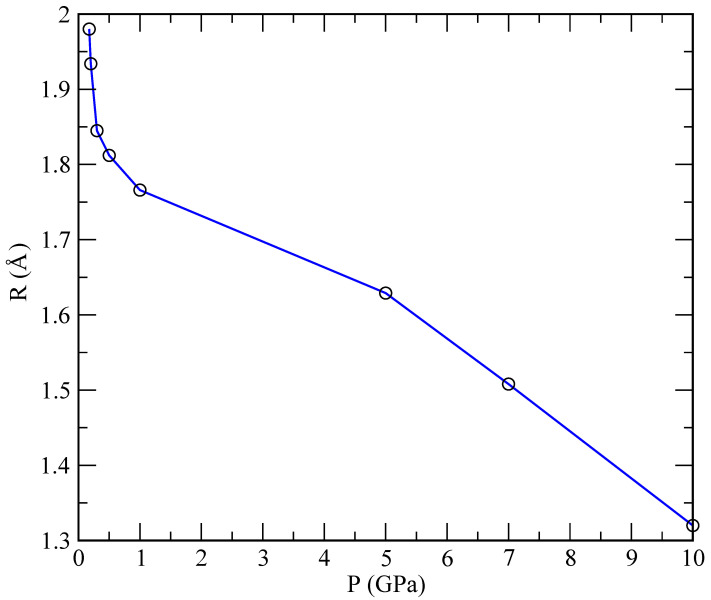
Radii of helium droplets for the pressure range 0.175–10 GPa.

**Figure 10 materials-15-02866-f010:**
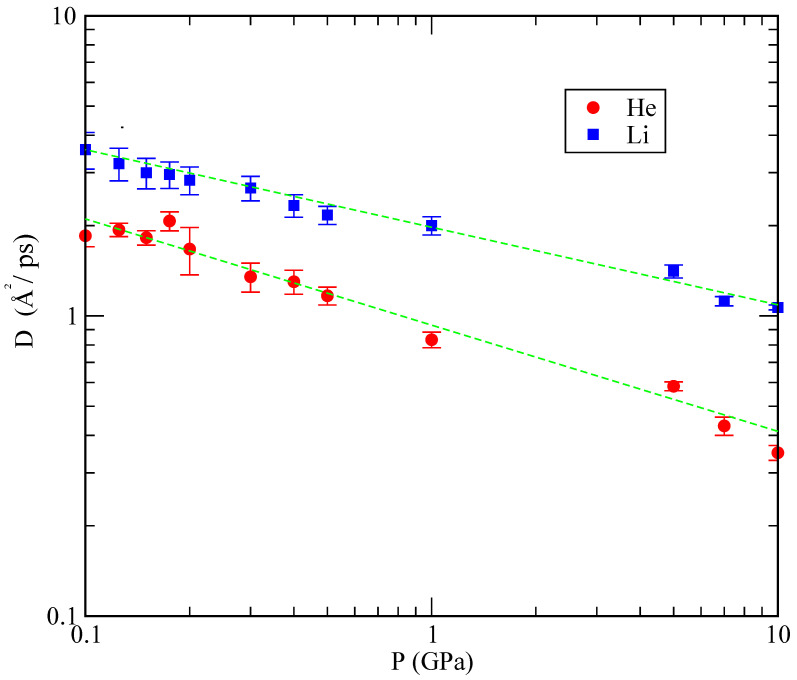
Diffusion coefficients of lithium (blue squares) and helium (red circles) at 800 K as a function of the pressure on a logarithmic scale. Green straight lines are a guide to the eye.

**Figure 11 materials-15-02866-f011:**
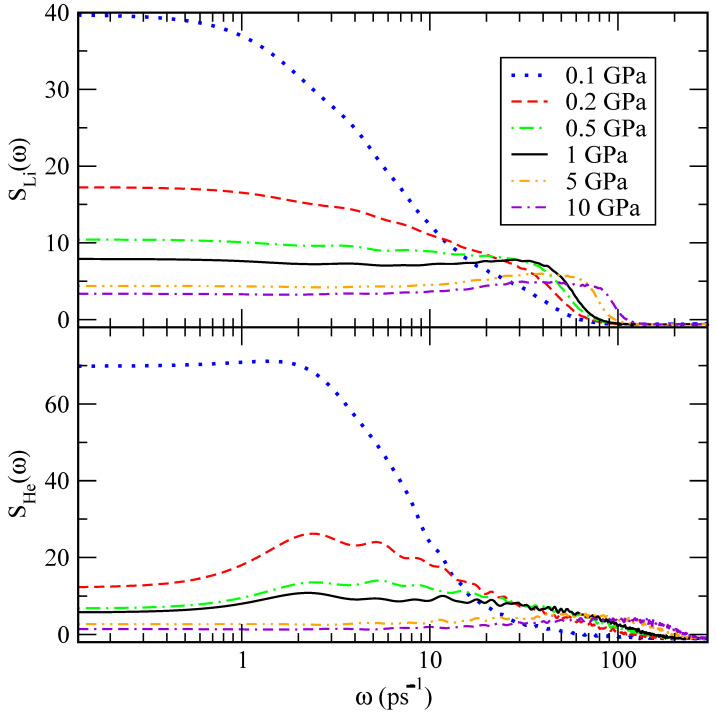
Spectral densities of lithium (**top**) and helium (**bottom**) at 843 K as a function of pressure. Lines: 0.1 GPa (dotted blue); 0.2 GPa (dashed red); 0.5 GPa (dot-dashed green); 1 GPa (black); 5 GPa (dot-dot-dashed orange); 10 GPa (dash-dash-dotted violet).

**Table 1 materials-15-02866-t001:** Average internal energies (*U*), pressures (*P*) and temperatures (*T*) for the simulated setups. All MC simulations considered $10^{8}$ sampling moves and all MD simulations were of total length 200 ps.

Method	*U*(K)	P (GPa)	T(K)
	71.1	0.104	843
	−187.0	0.122	843
	−430.3	0.146	843
	−553.3	0.178	843
MC	−700.0	0.208	843
	−1183.7	0.500	843
	−1398.6	1.005	843
	−1197.6	5.002	843
	−261.9	9.994	843
	150.8	0.105	842.3
	−214.2	0.128	842.1
	−380.4	0.152	842.1
	−507.6	0.177	842.1
MD	−720.8	0.202	841.9
	−1182.3	0.501	841.7
	−1405.2	0.999	841.6
	−1199.1	4.999	841.0
	−262.8	9.998	840.3

**Table 2 materials-15-02866-t002:** Binding energies of helium at 800 K as a function of the pressure.

Pressure (GPa)	$U_{\text{He-Li}}$ (K)	$U_{\mathrm{Li}}$ (K)	$U_{\mathrm{binding}}$ (K)
0.1	110	−58	4.2
1	−1402	−1413	0.275
10	−262	−188	−1.85

**Table 3 materials-15-02866-t003:** Radii of helium droplets at 843 K as a function of the pressure.

Pressure (GPa)	$R_{\mathrm{He}}$ (Å)
0.175	1.98
0.2	1.93
0.3	1.85
0.5	1.81
1	1.85
5	1.63
7	1.51
10	1.32

**Table 4 materials-15-02866-t004:** Diffusion coefficients of lithium and helium at 843 K as a function of the pressure.

Pressure (GPa)	$D_{\mathrm{He}}$ (Å${}^{2}$/ps)	$D_{\mathrm{Li}}$ (Å${}^{2}$/ps)
0.1	1.850	3.583
0.125	1.936	3.217
0.15	1.822	3.010
0.175	2.071	2.958
0.2	1.670	2.833
0.3	1.350	2.667
0.4	1.300	2.333
0.5	1.167	2.167
1	0.833	2.006
5	0.583	1.408
7	0.430	1.120
10	0.350	1.067

## Data Availability

Not applicable.

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
