# Peer review of "Nucleation of Helium in Liquid Lithium at 843 K and High Pressures"

_materials, 2022, doi:10.3390/ma15082866_

Round 1

Reviewer 1 Report

Future fusion reactors rely on tritium breeders (in the form of Li2SiO4/Li2TiO3 ceramic pebbles for solid blankets or Li-Pb eutectic for liquid blankets) to generate new fuel in order to achieve tritium self-sustainability. Helium, as the byproduct of the n-Li reaction, would especially have an impact on the physical, thermal, and tritium transport properties of the Li-Pb eutectic in the liquid blanket case, due to its low solubility. This has long been noticed as an issue that is not fully understood in fusion engineering. The work described in the current manuscript shows a first-step effort to simulate nucleation of He in liquid Li at fusion reactor operation conditions, based on classic nucleation theory, and using MC and MD methods. The theory and methodology are simple and nicely demonstrated, with sufficient details. The results provide enough significance and insights towards the full understanding of the He nucleation issue. I recommend the paper can be published in the present form.

Author Response

Thanks a lot for such positive answer. No related modifications, except we revised English spelling and wording.

Reviewer 2 Report

  1. Article must be structured as follow: Introduction (actuality etc.), Literature review, Problem formulation, Methods, Result and discussion, Conclusions. It is not logically to provide results before methods. There are no clear analysis of the current problem state.
  2. Which tools were used for the simulation?
  3. The self-citation must be decreased.
  4. It is not clear how results were verified

Author Response

We submit our response in the cover letter attached.

Reviewer 3 Report

Because of the high plasma temperature a pure deuterium nuclear fusion requires, only deuterium-tritium (D-T) fusion is practical.

The necessary tritium is here produced in a nuclear reaction from lithium-6 (eq. 2). This reaction also generates helium-4, which may outgas from the liquid metal solvent due to its low solubility at the operation temperature (843 K).

The authors basically investigated with Molecular Dynamics the consequences of the pressure dependence of the helium solubility in lithium melts and took into account non-ideal deviations from Henry's law by applying appropriately fitted pair potentials (eqs. 11-13).

This is an interesting contribution in terms of methodology and warrants publication.

However, the model would need significant extension before it can be of practical value.

First, the main reason for supersaturation is the dependence of the internal energy of the helium atoms in the droplets on surface tension. Smaller droplets have higher surface tension and therefore higher solubility. This should be stressed when discussing the critical droplet size.

Second, the lithium and helium can form van-der Waals molecules which were not mentioned in the text. A simple Lennard-Jones type of pair potential (eq. 13) is definitely too simple.

A more fundamental critique is due to the complete neglect of the chemistry of these mixtures. I have also not seen the effect of tritium presence mentioned in the paper.

First, the T atoms formed will soon react to give T2 molecules. Because T2 has a very limited solubility in lithium melts at 570 degree Celsius, their behaviour could be a game changer for the helium solubility. Besides, T2 reacts readily with lithium to give LiT. Moreover, T atoms themselves are even more reactive, giving the corresponding lithium hydride (LiT) almost immediately. Formation of dilithium (Li2) molecules (approximately several percent of the total) further complicates the picture. To make matters worse, the tritide anions could reduce lithium cations to give tritium cations (T+) which may react with the helium to produce helonium (HeT+) which is highly reactive itself (the strongest acid known) etc.  

I understand that the simulation of all these processes is complicated and justifies approximations. But at least their neglect should be mentioned in the text.

I can recommend acceptance of this paper after all of the above points had been addressed.

Author Response

(The authors gave the same response as above.)

Round 2

Reviewer 2 Report

none